# Highly Luminescent Copper Nanoclusters Stabilized by Ascorbic Acid for the Quantitative Detection of 4-Aminoazobenzene

**DOI:** 10.3390/nano10081531

**Published:** 2020-08-04

**Authors:** Qiang Li, Yunhao Li, Heguo Li, Xiaoshan Yan, Guolin Han, Feng Chen, Zhengwei Song, Jianqiao Zhang, Wen Fan, Changfeng Yi, Zushun Xu, Bien Tan, Wei Yan

**Affiliations:** 1Hubei Collaborative Innovation Center for Advanced Organic Chemical Materials, Ministry of Education, Key Laboratory of Green Preparation and Application for Functional Materials, Hubei Key Laboratory of Polymer Materials, School of Materials Science & Engineering, Hubei University, Wuhan 430062, China; lee110844693@hotmail.com (Q.L.); yunhao@stu.hubu.edu.cn (Y.L.); chenfeng@stu.hubu.edu.cn (F.C.); songkukus@163.com (Z.S.); fanwen@hubu.edu.cn (W.F.); changfengyi@hubu.edu.cn (C.Y.); zushunxu@hubu.edu.cn (Z.X.); 2State Key Laboratory of NBC Protection for Civilian, Research Institution of Chemical Defense, Beijing 100191, China; yanxiaoshan@gmail.com (X.Y.); glhan8282@sina.com (G.H.); 3Key Laboratory of Material Chemistry for Energy Conversion and Storage, Ministry of Education, Hubei Key Laboratory of Material Chemistry and Service Failure, School of Chemistry and Chemical Engineering, Huazhong University of Science and Technology, Wuhan 430074, China; jqzhang1991@163.com

**Keywords:** copper nanoclusters, fluorescence, detection, ascorbic acid

## Abstract

As one of the widely studied metal nanoclusters, the preparation of copper nanoclusters (Cu NCs) by a facile method with high fluorescence performance has been the interest of researchers. In this paper, a simple, green, clean, and time-saving chemical etching method was used to synthesize water-soluble Cu NCs using ascorbic acid (AA) as the reducing agent. The as-prepared Cu NCs showed strong green fluorescence (with a quantum yield as high as 33.6%) and high ion stability, and good antioxidant activity as well. The resultant Cu NCs were used for the detection of 4-aminoazobenzene (one of 24 kinds of prohibited textile compounds) in water with a minimum detection limit of 1.44 μM, which has good potential for fabric safety monitoring.

## 1. Introduction

Metal nanoclusters are relatively stable microscopic or submicroscopic aggregates composed of 10 to 100 metal atoms, molecules, or ions through physical or chemical binding forces [1,2,3,4,5,6]. Compared with materials of other sizes, metal nanoclusters have been widely studied because of their excellent photostability, strong photoluminescence, good biocompatibility, and unique quantum size effect. The physical and chemical properties of metal nanoclusters varied with the number of atoms contained [7,8,9,10,11,12,13,14]. Except for the extensively studied Au and Ag nanoclusters, Cu nanoclusters (Cu NCs) have received increasing attention due to their relatively cheap price and abundant resources [15], which have been broadly used in the fields of biological imaging [16,17,18], ion detection [19], chemical sensors [20,21], and catalysis [22]. There are two main methods for the synthesis of Cu NCs: the bottom–up method and the top–down method. The strategies of bottom–up for the synthesis of Cu NCs can be mostly divided into two approaches. The first way is the electrochemical method. In this method, the anode is consumed, providing a source of metal ions by going through anodic dissolution. Metal ions are reduced at the cathode for the generation of metal NCs. Noelia et al. prepared Cu NCs in a thermostated three-electrode electrochemical cell, using an aqueous tetrabutylammonium nitrate solution as the electrolyte, where the anode was a Cu sheet, the cathode was a platinum sheet, and the reference electrode was Ag/AgCl [23]. The other way is wet chemical reduction. Using folic acid as a reducing agent and stabilizer, Upashi et al. successfully prepared blue fluorescent Cu NCs specifically targeted to folate receptors over expressed cancer cells [24]. Besides preparing nanoclusters from metal ions, Cu NCs can be also produced in a top–down strategy by using the chemical etching method. Deng et al. used ammonia as the etchant and reduced non-fluorescent Cu NCs (average diameter 3.7 ± 0.5 nm) to blue fluorescent Cu NCs with a diameter of 1.2 ± 0.3 nm [25]. However, the use of strong reducing agents and their side effect, low quantum yield, restrict their further application in many fields. Therefore, it is still desirable to synthesis Cu NCs with high quantum yield by a reproducible, facile, economical, and nontoxic method.

In recent years, there have been many groups studying copper nanoclusters protected by ascorbic acid. Using ascorbic acid as a protective and reducing agent, WenJie Zhang et al. [26] prepared light yellow CuNCs and used them for the detection of picric acid. They stirred the copper precursor solution at 70 °C in a dark environment for 8 h, and the final detection limit of the as-prepared CuNCs was 0.98 μM. Zhifeng Cai et al. [27] used the same synthesis strategy and used it for the detection of quercetin, but they did not choose a dark environment. The final detection limit of the CuNCs was 0.19 nM. In our work, a top–down synthesis strategy was provided to produce CuNCs, using ascorbic acid as a protective and reducing agent. Firstly, copper nanoparticles were synthesized under a mild environment. Afterwards, under the etching action of NaOH solution, the non-fluorescent copper nanoparticles were converted into bright yellow green fluorescent copper nanoclusters with the quantum yield (QY) of 33.6%. The final copper nanoclusters were used for the detection of p-aminoazobenzene (p-AAB). Through the optimization of reaction conditions (incubation temperature, reaction time, concentration), the best preparation parameter of Cu NCs was acquired. The obtained Cu NCs were characterized by Fourier transform infrared (FTIR) spectra, X-ray diffraction (XRD), fluorescence spectroscopy, X-ray photoelectron spectroscopy (XPS), ultraviolet absorption spectroscopy, dynamic light scattering (DLS), and matrix-assisted laser desorption/ionization time-of-flight mass spectrometry (MALDI-TOF), and they were found to exhibit strong green fluorescence under the irradiation of ultraviolet light at 365 nm with a quantum yield as high as 33.6% and high ion stability, as well as good antioxidant activity.

4-Aminoazobenzene (also known as aniline yellow, p-AAB) is one of 24 kinds of prohibited textile aromatic amine compounds that have been proven to be carcinogenic. It is likely that it will enter the soil and water body due to the illegal discharge of crop-spraying pesticide, dye wastewater, or the waste liquid of organic synthesis experiments. Therefore, the detection of 4-nitrophenol is helpful to reduce the threat of p-AAB to our health. Traditional methods of detecting p-AAB include electrochemical detection, capillary gas chromatography, high-performance liquid chromatography, electrospray ionization source (ESI), liquid chromatography-mass spectrometry (LC-MS) method, and so on. However, the cumbersome instrumentation and sophisticated pre-treatment procedure limit their practicability. 

In the subsequent experiment, the as-obtained Cu NCs showed specific detection ability for p-AAB. Based on the internal filter effect (IFE), the quantitative detection of p-AAB in the aqueous phase was successfully achieved, and its minimum detection limit is 1.44 μM, which has good potential for fabric safety monitoring.

## 2. Experimental Section

### 2.1. Materials

#### 2.1.1. Reagents and Materials

Ultrapure water (18.2 MΩ) and a reaction container of heat collector constant temperature heating magnetic agitator DF-101S (GongYi City to China Instrument Co., LTD, Henan, China) were used throughout the experiment. *L*-Ascorbic acid (AA, 99.7%), H_2_O_2_ (30%), ethanol absolute (AR), sodium hydroxide (AR), hydrochloric acid (36–38%), sulfuric acid (98%), nitric acid (HNO_3_, 65–68%), and other organic solvents were purchased from National Medicines Corporation Ltd. of Beijing China. Cupric nitrate trihydrate (Cu(NO_3_)_2_·3H_2_O, AR), 4-aminoazobe-nzene (p-AAB, AR), 4-phenylazophenol (p-PAP, AR), 4-aminobiphenyl (p-BPA, AR), 4-chloroaniline (p-CA, AR), aniline (AN, AR), quinine sulfate dehydrate (QS, 98%), Basic Red 1 (Rh 6G, AR), and Methyl Orange (MO, AR) were from Aladdin, Wuhan, China. 

### 2.2. Methods

#### 2.2.1. Synthesis of Cu NCs@AA

The schematic diagram of Cu NCs synthesis and p-AAB detection is shown in Scheme 1. All of the glassware was washed with aqua regia and rinsed with ultrapure water before usage. In an optimal synthesis protocol, a 25 mL round-bottom flask was successively added with 500 μL Cu(NO_3_)_2_ solution (50 mM), 8.5 mL ultrapure water, and a certain amount of freshly prepared AA solution (the mixture was stirred at a constant temperature under a water bath for a period of time) was obtained. A certain amount of NaOH was slowly added to the obtained Cu nanocrystals solution (1 M); the solution gradually changed from transparent light yellow to translucent yellowish green, and then it quickly changed to turbid orange. The reaction solution was quickly transferred to an ice bath for cooling at about 5 min. Then, the cooled reaction solution was centrifuged at high speed (10,000 r/min) for 15 min to separate the yellow-green transparent supernatant; then, prepared Cu NCs@AA aqueous solution was frozen in −18 °C for follow-up experiments. Solid Cu NCs@AA was obtained by freeze-drying of its aqueous solution.

#### 2.2.2. Selectivity of Cu NCs Toward Carcinogenic Azo Dye p-AAB

The selectivity of Cu NCs@AA toward carcinogenic azo dye p-AAB was conducted as follows. Small volumes of different nitro compound solutions (in water), with the same concentration, were added to the Cu NCs@AA stock solution in water (0.02 g·mL^−1^), respectively. The final concentration of all the aromatic amine compounds was set to 0.2 mM, and their emissions were measured at the excitation wavelength of 393 nm to sense p-AAB. The procedures for detecting p-AAB in water was as follows. The freshly prepared aqueous Cu NCs@AA stock solution was mixed with different concentrations of p-AAB dissolved in water (0–120 µM) at a 1:1 volume ratio. The mixed solutions were measured using a PerkinElmer LS55 fluorescence spectrometer at an excitation wavelength of 393 nm. The freeze-dried Cu NCs@AA powder was dissolved in water, the final concentration was set to 0.02 g·mL^−1^, and then, it was incubated with different concentrations of p-AAB dissolved in water (0–120 µM), at a 1:1 volume ratio. The fluorescence emission spectra were obtained at 393 nm excitation wavelength. The limit of detection (LOD) is based on 3 σ/K, where σ represents the standard deviation and K is the absolute value of the slope of the curve. K*sv* (quenching constant) is obtained using Stern–Volmer plots, F_0_/F = K*sv* [*c*] + 1, where F_0_ and F stand for the initial fluorescence intensity of Cu NCs and after adding p-AAB, respectively, and c is the concentration of the analyte. 

#### 2.2.3. Characterization

Fluorescence spectra were obtained with a LS-55 fluorescence spectrometer (PerkinElmer, Shanghai, China). UV-vis light absorption spectra were obtained using a UV-1800PC spectrophotometer (MAPADA, Shanghai, China). Transmission electron microscopy (TEM) images were obtained on a Tecnai G20 (FEI, Shanghai, China). Field emission scanning electron microscopy (FSEM) mapping results were obtained using a Sigma500 (Zeiss, Shanghai, China). Time-correlated single photon counting was performed with a FLS-980 luminescence spectrometer (Edinburgh Instruments, Shanghai, China). X-ray diffraction (XRD) measurements were performed by using a D8 Advance X-ray diffractometer (Bruker, Germany). FTIR spectra were obtained using a Nicolet iS50 Fourier transform infrared spectrometer (Thermo Fisher, Shanghai, China). X-ray photoelectron spectra (XPS) were obtained using an ESCALAB 250Xi (Thermo Fisher, Shanghai, China). Mass spectra were obtained by using a 5800MALDI TOF mass spectrometer (AB SCIEX, Framingham, MA, USA) in positive mode with trans-2-[3-(4-tert-butylphenyl)-2-methyl-2-propenylene]malononitrile (DCTB) as a matrix. The quantum yield (QY) of the as-prepared Cu NCs was calculated by the slope method, with Rhodamine 6G as a reference (QY = 0.95 in ethanol). Five different concentrations of Cu NCs in water and Rhodamine 6G in ethanol were prepared. Then, the optical density from UV-vis (O.D. < 0.05, to reduce the inner filter effect) and the fluorescence spectrum of each solution were obtained at the optimal excitation wavelength of 393 nm. The quantum yield of Cu NCs was calculated from the equation *Q* = *Q_R_* [*m*/*m_R_*] [*n_2_*/*n_R2_*] [13,28], where *Q* represents the quantum yield, *m* is the slope of the fitting line, subscript *R* is the reference of the known quantum yield, and *n* is the refractive index of solvent. All of the measurements were conducted at ambient temperature, if not noted otherwise.

## 3. Results and Discussion

### 3.1. Synthesis and Properties of Cu NCs@AA

Firstly, the non-fluorescent Cu nanocrystals were obtained by mixing an aqueous mixture of AA and Cu^2+^ precursors under mild conditions. AA is a green and mild reducing agent, which was used as both reducing agent and protective agent to react with the Cu^2+^ ions. After the addition of etching agent NaOH (1 M), copper nanocrystals could be quickly decomposed into small nanoclusters with fluorescence properties. The color of solution gradually changed from transparent yellowish to translucent yellowish green, and it soon changed to turbid orange. The process was about 5 min, and the reaction solution was quickly transferred to an ice bath. Then, the cooled reaction solution was centrifuged at high speed (10,000 r/min) for 15 min, to separate the yellow-green transparent supernatant, and Cu NCs@AA aqueous solution was obtained. Then, the aqueous solution was frozen under −18 °C for follow-up experiments. Solid Cu NCs@AA was obtained by freeze-drying its aqueous solution. A strong green fluorescence of the aqueous solution was observed under 365 nm irradiation, implying the formation of Cu NCs. The optimal excitation and emission wavelengths of the as-prepared Cu NCs were 393 nm and 505 nm, respectively (Figure 1a). The maximum emission peak of Cu NCs exhibited excitation independent behavior (Figure 1b), implying the relatively uniform surface states and that the emission is a real fluorescence from the relaxed state. No localized surface plasmon resonance peak appeared in the range of 500–600 nm in the UV-vis absorption spectrum (Figure 1c), which indicated the absence of larger copper nanoparticles. The average lifetime of Cu NCs@AA in water was determined to be 0.772 μs (99.99%) and 8.01 μs (0.01%) (Figure 1d) based on the single exponential function F(*t*) = 3.124 × 10^9^ e^(−*t1*/0.772)^ + 3020e^(−*t2*/8.01)^ + 0.7, x^2^ = 0.998, illustrating that the photoluminescence from the Cu core occurs through a pathway that only contains the first singlet state. The freeze-dried Cu NCs powder also displayed a strong green fluorescence under a UV lamp.

### 3.2. Characterization of Cu NCs@AA

Fourier transform infrared (FTIR) spectroscopy and Raman spectroscopy were used to characterize the characteristic structure of fluorescent Cu NCs protected by AA. The infrared spectra of AA and Cu NCs@AA are shown in Figure 2a. In the black line of AA, the strong absorption of 1754 cm^−1^ is caused by the stretching of Coluo carbonyl groups in the five-membered lactone ring system; here, the strong absorption of 1674 cm^−1^ is caused by the stretching of the enol-type C double bond, the strong absorption at 1320 cm^−1^ is caused by the bending vibration of enol OH, and 1277 cm^−1^ belongs to C–O–C stretching. The absorption in the 1046–1801 cm^−1^ region is attributed to the C–O–H bending on the branched chain structure, and the absorption in the 1027 cm^−1^ region is related to the deformation of the lactone ring. In the red line of Cu NCs@AA, it is obvious that the strong absorption band of the C double bond disappears at 1674 cm^−1^, the strong band of bending vibration of enol OH disappears at 1320 cm^−1^, and the absorption peak of conjugated Coluo appears at 1637 cm^−1^. The absorption peak of lactone Callio is red-shifted from 1754 cm^−1^ to 1733 cm^−1^, and the intensity is obviously weakened, indicating that AA is oxidized [29]. At the same time, it shows that the reduced Cu is coordinated with lactone, which is not coordinated with the ketone Curoo group converted by enol.

Raman spectra were used to assist the FTIR spectra with analyzing the structure of substances. Figure 2b recorded the Raman spectra of both Cu NCs@AA and AA solids in the droplets. The presence of water did not interfere with the Raman test, because the Raman peak intensity of the water was very low. The small scattering peak of 1753 cm^−1^ came from the Czocho stretching of AA, and the peak of 1661 cm^−1^ belonged to the Czochralski double bond stretching of AA. In the Raman band of Cu NCs@AA (Figure 2b), the wide scattering band at 445 cm^−1^ was the stretching of the Cu–O bond, which further confirmed the binding of the Cu atom to O coordination in AA.

In order to obtain the information of the valence and bonding properties of metal nanoclusters, X-ray photoelectron spectroscopy (XPS) is a very accurate and precise research technique. In Figure 3a, the Cu 2p level had two characteristic 2p3/2 and 2p1/2 splitting components, with spikes at about 932.9 eV and 952.8 eV, respectively, corresponding to zero-valent copper.

However, because the difference of 2p_3/2_ binding energy between Cu (0) and Cu (I) is only 0.1 eV, around 932 eV, it was still difficult to distinguish Cu (0) and Cu (I) accurately. In addition, there was no signal corresponding to the CuO oscillating satellite, which clearly indicated that Cu (II) was not in the system [30]. In metal nanoclusters, each metal atom is important to its structure and photo physical properties. It was also obtained from the XPS full spectrum and the element content table (the ratio of the Cu element to the C element in Figure 3b) that Cu NCs@AA was about 1:5. Next, in the charge-corrected C 1s and O 1s comparison images (Figure 3c,d), we could see that the π–π* satellite peak of the Cure C double bond (about 291.6 eV) disappeared in Cu NCs@AA, indicating that dehydroascorbic acid was bound to the surface of Cu NCs. At the same time, the binding energy positions of C 1s and O 1s of Cu NCs@AA moved to the lower level compared with AA. In the O 1s spectrum, the binding energy of Cmuro and Coluo migrated to the lower position up to 0.5 eV, indicating the formation of a Cu–O (coordination) bond. In addition, the charge transfer (LMMCT) from ligand to Cu–Cu and then to ligand resulted in the strong luminescence of Cu NCs@AA.

Transmission electron microscope (TEM) provided strong evidence from the point of view of size, morphology, and electron diffraction analysis to prove whether metal nanoclusters are formed or not. The samples of Cu NCs@AA obtained under the best conditions and etching was analyzed by TEM, and the results are shown in Figure 4. Before etching, it was obvious that the Cu nanocrystals stabled by AA are spherical and the size distribution is uniform with an average particle size of 4.95 ± 0.92 nm (Figure 4a1,b). After NaOH treatment, the original large Cu nanocrystals were decomposed into small NCs with an average diameter of 1.59 ± 0.23 nm (Figure 4c1,d). Figure 4e showed a typical image of coexistence in the Cu nanocrystals (red circle) and the Cu NCs (yellow circle) during the etching process. TEM analysis showed that the nuclear size of all the Cu nanocrystal particles was less than 2 nm after etching. Therefore, Cu nanoclusters were successfully prepared by chemical etching.

In addition, the field emission scanning electron microscopy mapping showed that the related Cu, C, and O elements were uniformly distributed in the clusters in the Cu NCs@AA solid powder (Figure 5a,b). Furthermore, the selected surface elements of the samples were analyzed by an X-ray energy dispersive spectrometer, and the composition elements of the samples were Cu, C, and O, which were the same as those of XPS, and the proportion of Cu elements was 7.1%.

Matrix-assisted laser desorption/ionization time-of-flight mass spectrometry (MALDI-TOF MS) had obvious advantages in determining the composition and structure of macromolecular compounds such as metal nanoclusters. According to the MALDI-TOF mass spectrum data of Cu NCs@AA (Figure 5c), the interval between the adjacent main peaks was about 44 (m/z), which is exactly equal to 1/4 of the molecular weight of AA, and some of the main peaks could be assigned to the corresponding attribution, for example, the mass–nucleus ratios (m/z) are 1041.69, 1085.70, and 1173.76, which can be attributed to [Cu_5_L_4_+H_2_O+H]^+^, [Cu_6_L_4_]^+^, and [Cu_7_L_4_+Na+H]^+^, respectively (L stands for AA (C_6_H_8_O_6_)). Mass spectrometric analysis showed that the Cu NCs@AA core was composed of 5–7 copper atoms, and the outside of the copper core was protected by 4 AA ligands. This structure was similar to the green fluorescent Cu NCs protected by L-cysteine prepared by the Amitava team [31], which proved the rationality of the Cu NCs@AA structure. At the same time, it reflected the size-dependent fluorescence properties of the metal nanoclusters.

### 3.3. The Effects of Reaction Conditions on Cu NCs@AA

The mixing temperature of Cu nanocrystals is the most important parameter in the preparation of Cu NCs. The results of temperature optimization during mixing are shown in Figure 6a,b. The fluorescence spectra showed that the final clusters have the strongest fluorescence at the temperature of 45 °C. The number of Cu nanocrystals obtained at lower temperature was limited, while Cu nanocrystals grew too fast at higher mixing temperature; both of them could not be conducive to the final fluorescence performance of Cu NCs. Arun Chattopadhyay’s team also used the optimum mixing temperature of 45 °C in the preparation of lysozyme template-protected Cu NCs [32]. In addition, it was worth noting that when the incubation temperature was as high as 65 °C, the spectrum showed a certain blue shift, indicating that the structure of the cluster changed to a certain extent.

The mixing time of Cu nanocrystals also had a great influence on the fluorescence properties of the final copper nanoclusters. The time optimization data of the mixing process are shown in Figure 6c,d. When the incubation time is 2.5 h, the fluorescence intensity of the clusters is the strongest, while further prolonging the heating time leads to the decrease of the fluorescence intensity of the clusters. The longer incubation time was good for the reduction reaction and perfect structure of Cu nanocrystals. When the mixing time jumped over the optimal point, the fluorescence intensity began to decrease, which might be due to the uneven etching of Cu nanocrystals due to excessive growth or larger aggregation, which led to the decrease of fluorescence intensity. The Cu nanocrystals had almost no fluorescence effect, as shown in the curve of an NaOH dosage of 0 μL in Figure 6e,f, while the etching agent NaOH can decompose Cu nanocrystals into Cu NCs and emit strong green fluorescence.

In this process, NaOH played two roles: enhancing the reduction ability of AA (deprotonation) and etching Cu atoms on the surface of nanocrystals. The results of optimizing the amount of NaOH during etching are shown in Figure 6e,f. With the increase of the amount of NaOH, the fluorescence intensity of the corresponding clusters increased significantly. When the addition amount was more than 200 μL, the fluorescence intensity decreases greatly, which was due to the excessive deprotonation of AA caused by the high concentration of NaOH, leading to the decrease of its stability. Therefore, the optimum dosage of etching agent NaOH (1M) was 200 μL, and the final concentration is 11.8 mM. The pH of the corresponding solution was about 8–9. This result was the same as that reported in the literature, the weak alkaline condition of pH = 8–10 was more favorable for the etching reaction of Cu nanocrystals.

In this method, AA acted as both reducing agent and protective agent, so its dosage would have a great impact on the fluorescence properties of the Cu NCs@AA. The excessive addition of AA would make the reduction rate of Cu^2+^ too fast, which would lead to the passivation of Cu nanocrystals, increasing defects, a deterioration of dispersion, and the final formation of large nanoparticles [33]. When the amount of AA was too small, the degree of reduction was incomplete, which would lead to low stability and an agglomeration of Cu nanocrystals. Under the above conditions, the effect of the amount of AA on the fluorescence performance of Cu NCs@AA was further discussed. The experimental results are shown in Figure 6g,h. It was known that the optimum addition of AA (100 mM) was 2.0 mL, and the final concentration was 11.8 nM.

### 3.4. Stability of Cu NCs@AA

As a special zero-dimensional nanomaterial, whether metal nanoclusters can ensure the stability of fluorescence properties is undoubtedly one of the key factors before they are put into application. The stability of Cu NCs@AA mainly included time stability, ionic strength stability, and antioxidant stability. The characterization results are shown in Figure 7. The newly prepared Cu NCs solution (0.4 mL) was added with different concentrations of 0.8 mL NaCl solution, which had an effect on the fluorescence of Cu NCs, and the fluorescence of NCs decreased gradually with the increase of NaCl concentration (Figure 7a). When the concentration of NaCl was 100 mM, the strength of PL decreased by about 50%, indicating that the stability of ionic strength of the prepared Cu NCs was not ideal. From the results of Figure 7b, it could be seen that Cu NCs showed good antioxidant stability to lower concentrations of H_2_O_2_. In addition, the solution still maintained about 91% PL strength after being frozen away from light for 7 months (Figure 7c).

The effects of 14 common metal ions Na^+^, K^+^, Ca^2+^, Mg^2+^, Al^3+^, Zn^2+^, Mn^2+^, Co^2+^, Ni^2+^, Cu^2+^, Ag^+^, Pb^2+^, Fe^2+^, and Fe^3+^ on the fluorescence properties of Cu NCs@AA were also discussed (Figure 8a). The newly prepared Cu NCs@AA aqueous solution was mixed with the nitro compound solution of these 14 metal ions (100 mM) at a 2:1 volume ratio. The normalized PL results are shown in Figure 8. It could be seen that Ca^2+^, Al^3+^, Zn^2+^, Mn^2+^, Co^2+^, Ni^2+^, Cu^2+^, Ag^+^, Pb^2+^, Fe^2+^, and Fe^3+^ could quench the fluorescence intensity of Cu NCs, but only Cu^2+^, Ag^+^, Pb^2+^, Fe^2+^, and Fe^3+^ had the most obvious quenching effect. The possible reason for analyzing the quenching mechanism was that these transition metal ions form a strong coordination with O (including branched chain OH or lactone Czochro) in AA molecules, resulting in Cu NCs@AA’s aggregation-induced fluorescence quenching [34] or the protective agent’s detachment from the copper core surface, and fluorescence “turn-off” induced by a charging transfer between the ligands and metal ions [3].

### 3.5. The Selectivity for Detection of p-AAB

4-Aminoazobenzene (also known as aniline yellow, p-AAB) is one of 24 kinds of prohibited textile compounds and has been proven to be carcinogenic. In the International Environmental Textile Association’s “Ecological Textile Standard 100” and GB/T 18885-2009 “Ecological Textile Technical requirements”, p-AAB is on the list of 24 banned aromatic amine compounds in textiles. It is likely that it will enter the soil and water body due to the illegal discharge of crop-spraying pesticide, dye wastewater, or the waste liquid of organic synthesis experiments [35]. Therefore, the detection of 4-nitrophenol is helpful to reduce the threat of p-AAB to our health. Traditional methods of detecting p-AAB include electrochemical detection, capillary gas chromatography, high-performance liquid chromatography, ESI source, LC-MC method, and so on [36]. However, the cumbersome instrumentation and sophisticated pre-treatment procedure limit their practicability.

The selectivity and sensitivity of Cu NCs@AA in detecting p-AAB were studied (Figure 8b). The first was selective exploration. The same concentration of 4-chloroaniline (p-CA), 2-toluidine (o-TD), 4-chloro-4-diaminodiphenyl ether (ODA), 4-aminobiphenyl (p-BPA), aniline (AN), 2-dinitroaniline (DTA), 4-hydroxyazobenzene (p-PAP), and p-AAB were mixed with the same volume of Cu NCs solution at room temperature, and PL spectra were measured. It could be seen from Figure 8b that Cu NCs had a good selectivity for the detection of p-AAB, and the quenching of p-AAB reached 82% under the same conditions. Other aniline compounds, such as p-BPA, p-CA, ODA, and o-TD are also on the list of 24 banned aromatic amines in textiles, but these four compounds have a weak effect on the fluorescence of Cu NCs. Among the other three non-textile banned aromatic amines p-PAP, DTA, and AN, only DTA showed a strong PL quenching effect, except for p-AAB, which reached 21% under the same conditions. This result can still show that the Cu NCs have good selectivity for the detection of p-AAB.

After determining the selectivity, the sensitivity of Cu NCs for detecting p-AAB was further evaluated (Figure 9a,b). As shown in the digital photo (Figure 9e), with the gradual increase of p-AAB concentration from left to right, it was obvious that the color of the Cu NCs solution deepens gradually under sunlight, and the fluorescence intensity of the Cu NCs solution decreases gradually under ultraviolet light, which could be distinguished by the naked eye. The fluorescence of the sample was tested, and the standard curve was obtained. From the PL spectral data, the addition of p-AAB did not change the position of the Cu NCs emission peak, and the intensity of the adjacent spectra changed obviously. From the fitting of PL peak data (Figure 9c), there was a good linear relationship between the concentration of p-AAB and the corresponding PL peak intensity in the concentration range of 0–16 μM (*R^2^* = 0.990). The linear fitting equation is *Y* = −16.97*X* + 722.97. The LOD (limit of detection) of Cu NCs to p-AAB is 1.44 μM (defined as 3 σ/K), with a relative standard deviation σ = 8.1%. This limit is much lower than that of the latest EU standard ISO 14362-3: 2017 “Determined levels of 4-aminoazobenzene” (≤30 mg/kg) and the national mandatory standard GB 18401-2010 stipulates “decomposable aromatic amine dyes” (≤20 mg/kg) [37].

Quenching constant is a parameter to measure the effect of fluorescence quenching. According to the Stern–Volmer equation, the fitting straight line *F_0_*/*F_max_* = 0.0397 [*c*] + 0.9528 was obtained, and the quenching constant K_sv_ of p-AAB to Cu NCs in aqueous solution was calculated to be 3.97 × 10^4^ M^−1^ (Figure 9d).

The mechanism of fluorescence quenching of Cu NCs caused by p-AAB was analyzed. Basically, the fluorescence quenching process could be divided into two different mechanisms, a static quenching mechanism (ground state complex formation) and a dynamic quenching mechanism (diffusion collision) [38]. In general, there are many possible mechanisms that contribute to the quenching of fluorescence emission from nanomaterials, such as fluorescence resonance energy transfer (Forster resonance energy transfer, FRET), internal filtering effect (inner filter effect, IFE) and electron transfer (electron transfer, ET) [39]. In order to understand the exact mechanism of Cu NCs fluorescence quenching mediated by azo dye p-AAB, the fluorescence lifetime of azo dye was tested (Figure 10a). After the addition of p-AAB, the average fluorescence lifetime of Cu NCs (0.746 μs) was slightly lower than that before p-AAB (0.772 μs), indicating that p-AAB may inhibit the fluorescence of Cu NCs mainly through a static quenching mechanism. By comparing the spectral superposition (Figure 10b), we can see that there is the most overlap between the absorption spectrum of p-AAB and the excitation spectrum of Cu NCs, indicating that the fluorescence inhibition of Cu NCs is mainly from the IFE, which also explains the high selectivity of p-AAB detection. Moreover, due to the wide absorption spectrum of p-AAB, it also had a small overlap with the emission spectrum of Cu NCs@AA, which might lead to the FRET effect. From the results of the slight decrease of fluorescence lifetime, it was found that there is a small part of the dynamic quenching mechanism in the p-AAB/Cu NCs@AA system, which is consistent with the inhibition of fluorescence by FRET. It is worth noting that the existence of the FRET mechanism helps to improve the sensitivity of detection.

## 4. Conclusions

In summary, using ascorbic acid as a reducing agent and protective agent, strong green fluorescent copper nanoclusters (Cu NCs@AA) were simply and effectively prepared in aqueous solution by the etching method with a high quantum yield over 33.6%. The as-prepared Cu NCs showed a decent response for the selective detection of p-AAB in aqueous media, with an acceptable sensitivity. This study broadens the scope of applications of Cu NCs for the detection of TNP, with decent potential for the monitoring of environmental security.

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
