# Peer review of "Highly Luminescent Copper Nanoclusters Stabilized by Ascorbic Acid for the Quantitative Detection of 4-Aminoazobenzene"

_nanomaterials, 2020, doi:10.3390/nano10081531_

Round 1

Reviewer 1 Report

I quite like the idea of this manuscript which is to use luminescent copper nanoclusters as a mechanism to quantitatively monitor and detect the presence of 4-aminoazobenzene. Nonetheless, there are a number of minor issues, particularly in the introduction, that reduce my overall enthusiasm for the manuscript. I recommend that the following issues be corrected before the manuscript be accepted for publication:

  1. Overall, the manuscript has some syntax-based errors that detract from the ability to focus exclusively on the scientific content of the manuscript. A thorough proofreading to address this issue is recommended.
  2. The authors use a fairly broad definition of metal nanoclusters as “microscopic or submicroscopic” and containing “several or even thousands of atoms.” This is a broad range and properties are expected to be different within that range – i.e. a nanocluster with several atoms will behave differently than a nanocluster that contains thousands of atoms. The authors should consider narrowing their definition and/or differentiating between different sizes of nanoclusters within these ranges.
  3. The authors characterize this method as “green.” They need to clarify what parts of their synthetic methods is considered “green” and how this general adjective is applied.
  4. The authors broadly claim that traditional methods for detecting p-AAB involve “cumbersome instrumentation” and “sophisticated pre-treatment procedure.” This is a broad characterization and needs to be described in more precise detail if they are claiming substantial improvements for their method compared to what currently exists.

Reviewer 2 Report

The manuscript "Highly luminescent copper nanoclusters stabilized by ascorbic acid for the quantitative detection of 4-aminoazobenzene", written by Li et al., describes a synthesis and application of Cu nanoclusters in a detection of 4-aminoazobenzene. The manuscript deals with a relatively interesting topic. However, I do have some comments before I will be able to recommend this paper.

  1. Lines 288-305: Final amount of NaOH should be given in a concentration rather than in absolute amount in microliters. The same applies also to ascorbic acid.
  2. How do authors explain the slight blue shift in the spectrum at 7C after 7 months of storage?
  3. Line 338: Phase textile prohibited is not adequate and it should be rephrased.
  4. Description of the shown mass spectra is relatively shallow. Explanation of the mass differences should be extended. If there is Cu7L4+Na+H, there should also be Cu6L4+Na+H; Cu5L4+Na+H etc. Why is there a difference of 1/4L? What about the charge of the described species?
  5. Scale-bars in the figure 4 do not fit with the text description.
